# Healthy Behaviors through Behavioral Design–Obesity Prevention

**DOI:** 10.3390/ijerph17145049

**Published:** 2020-07-14

**Authors:** LesLee Funderburk, Thomas Cardaci, Andrew Fink, Keyanna Taylor, Jane Rohde, Debra Harris

**Affiliations:** 1Robbins College of Health and Human Sciences, Baylor University, Waco, TX 76798, USA; Tom_cardaci1@baylor.edu (T.C.); Andrew_fink@baylor.edu (A.F.); keyanna_taylor@baylor.edu (K.T.); debra@rad-consultants.com (D.H.); 2JSR Associates, Catonsville, MD 21228, USA; jane@jsrassociates.net

**Keywords:** obesity, built environment, nutrition, physical activity, indoor environment quality

## Abstract

Evidence for behavior modification for improved health outcomes was evaluated for nutrition, physical activity (PA), and indoor environmental quality (IEQ). The databases searched included LISTA, PubMed, and Web of Science, with articles rated using an a priori baseline score of 70/100 to establish inclusion. The initial search produced 52,847 articles, 63 of which were included in the qualitative synthesis. Thirteen articles met inclusion for nutrition: cafeteria interventions, single interventions, and vending interventions. Seventeen articles on physical activity were included: stair use, walking, and adjustable desks. For IEQ, 33 articles met inclusion: circadian disruption, view and natural light, and artificial light. A narrative synthesis was used to find meaningful connections across interventions with evidence contributing to health improvements. Commonalities throughout the nutrition studies included choice architecture, increasing the availability of healthy food items, and point-of-purchase food labeling. Interventions that promoted PA included stair use, sit/stand furniture, workplace exercise facilities and walking. Exposure to natural light and views of natural elements were found to increase PA and improve sleep quality. Overexposure to artificial light may cause circadian disruption, suppressing melatonin and increasing risks of cancers. Overall, design that encourages healthy behaviors may lower risks associated with chronic disease.

## 1. Introduction

The obesity epidemic in the United States (U.S.) requires a response from a public health perspective, emphasizing the changes in environmental drivers that influence the personal, social and economic health of society [1]. The environmental driver of the built environment is an important avenue to consider regarding the behavior modification that can influence the overall health. Active Design is a contemporary design concept that is meant to address those features of the built environment that have been shown to support healthy eating and routine physical activity (PA) [2]. Design elements that influence the physical space to make it easier for individuals to engage in healthier choices as the norm and make less healthy choices more difficult are key [3,4]. 

This literature review is guided by Florence Nightingale’s environmental theory that was informed by her experiences over the course of her adult life of service. Nightingale employed a pragmatist’s approach for practical action to remedy human suffering [5]. Nightingale’s environmental theory can be viewed as a systems model that focuses on the “person” in the center, surrounded by aspects of the environment, all in balance. Her environmental theory [6] addresses the influence of several elements including fresh air and ventilation, thermal comfort, natural light and nutrition, as factors working together to provide a supportive environment for health. The practical approach becomes action to implement the changes necessary to create balance for the person, supportive of complete health. 

The same environmental factors influence the aspects of the human condition through the integration theory, explained as a group of models assimilated to understand the complexity of the human-environment relationship [7]. Within the integration theory, the transactional model is responsive to the conceptual framework of Florence Nightingale’s environmental theory. The transactional model asserts that the human-environment relationship is mutually supportive so that the environment affects the user just as the user affects the environment [8], which lends to a holistic system for balancing the human-environment relationship. In public health, the aim of environmental theory is to improve health and reduce health risks, by providing environmental conditions that passively or directly influence persons to take actions to improve their own health [9].

A health condition affecting two-thirds of U.S. adults is overweight and obesity [10], with obesity rates currently at 42.4% [11]. Body mass index (BMI) is commonly used to estimate overweight or obesity and is defined as the weight in kilograms divided by the height in meters squared [10]. A person with a BMI of 25.0–29.9 is considered overweight while those with a BMI over 30.0 are considered obese [10]. Of further concern is that being overweight/obese is a risk factor for type 2 diabetes mellitus (T2DM), with rates of T2DM and prediabetes rising across the U.S., effecting more than 30 million and 84 million people, respectively [12]. Evidence suggests that with the proper lifestyle changes of improved nutritional intake and increased physical activity (PA), both obesity and T2DM are preventable diseases [13,14,15]. Therefore, effective strategies for improving nutritional intake and increasing PA are needed to combat this pandemic. Despite the knowledge that dietary modifications and PA [13,14,15] will improve an individual’s overall health, there have been substantial increases in the rates of obesity and T2DM diagnosed across the world [16,17].

The human-environment relationship is complex and requires a comprehensive method to create transformational change in health. Indoor environmental quality (IEQ) influences how occupants interact with the built environment. When designing the built environment, decisions are made that contribute to the interaction between the environment and the occupant, where the environment directly or indirectly affects occupant behavior. This behavioral change may be active, for example, the layout of an office, which affects traffic patterns; or the behavioral change may be passive. Passive compliance in the built environment may be direct or indirect, but is experienced passively, accepting the exposure to the environmental condition without active response or resistance; for example, office design with exposure to natural light, affecting human circadian entrainment, which influences sleep and mood. 

Since Americans spend a significant amount of time at the workplace, this is an ideal environment to promote positive lifestyle activities and changes [18]. Strategies to increase compliance to make positive choices while at work regarding eating habits and PA are crucial. It is well established that the built environment influences the worker’s experience and behavior and in turn effects the return on investment to occupant organizations, regarding productivity, reduced absenteeism, and reduced health care costs [19,20,21]. Workplace environments actively designed for health may provide support for healthy behaviors. This literature review focuses on nutrition, PA, and IEQ, specifically lighting and views, to explore the connections between the built environment and behavior modification leading to healthy outcomes for the individual.

## 2. Materials and Methods 

The literature search was performed using One Search to search across all digital subscribed databases, including LISTA, PubMed, and Web of Science. Peer-reviewed published research, available through online sources, literature reviews, meta-analyses, and other sources were evaluated for inclusion. The search terms are shown in Table 1. The articles were then scored using a peer-reviewed rating tool [22] utilizing a benchmark score of 70 points or higher for inclusion in the review.

The peer-reviewed article rating tool used in this study was validated using Cohen’s Kappa Weighted to measure the agreement between two raters for inter-rater reliability [23]. The rating system process uses search parameters, including identifying databases for inclusion, search terms, and sources of evidence. Specific inclusion criteria were developed prior to the full-text assessment to determine eligibility. Inclusion criteria focused on the behavioral, medical, and health outcomes related to nutrition and physical activity that were found to be direct influences that may be affected by the selected IEQ factors, presenting direct and indirect influences on the outcomes. For instance, IEQ factors that affect sleep quality may contribute to increased consumption and increased weight. Articles in English involving human subjects related to the built environment and the programs focused on nutrition, physical activity, and indoor environmental quality, specifically lighting and views, were included. Articles were excluded if not specifically related to the built environment and programs focused on nutrition, physical activity and lighting and views. Other exclusion criteria were based on methodological limitations (statistical power, validity, and reliability), effecting the quality of the results presented. A critical analysis of the published research using a structured method was performed [24,25,26,27].

The article rating system is based on two prerequisites and a 100-point score assigned across six major study design levels. The maximum possible points allocated to each study design level are weighted based on the literature, which provides an orderly approach for qualifying the evidence [25,26,28]. After an article was evaluated, a predetermined designated baseline score of 70 established inclusion for the systematic literature review. Using the article rating system tool, the data from articles that met inclusion were extracted for the evaluation of study characteristics, participant characteristics, intervention, setting, and results. Specifically, the two prerequisites focused on the source of evidence and external bias. The article must be published in a peer-review journal, credible report, or academic dissertations and theses. External funding and conflicts of interest must be addressed. Research designs were grouped into six levels ranging from the meta-analysis of multiple randomized controlled experiments (level 1) to evidence based on expert opinion (level 6). Levels 2–5 included experimental studies, systematic and integrative reviews, quasi-experimental studies, exploratory studies, and case studies, respectively. First, the articles were assessed by title and abstract; then, they were evaluated based on the full text. Once the research design was determined, the article was assessed for the rationale and objectives, methods (ethical considerations, hypotheses, sampling, variables, data collection, data analysis, reliability, and internal and external validity). The results were evaluated based on the methods presented. The discussion and conclusions were assessed for relevance, the interpretation of the findings, and the use of evidence to support the significance of the work. Limitations are also part of the review rating system and focus on the limitations of the methods (sample size, confounding variables, and the variables that cannot be controlled). Recommendations should address the practical application and future research that are formed from the outcomes of the study. The reference management software Zotero and EndNote X9 were used to manage the searches and write the review.

Among the 52,847 articles acquired during the search, 52,354 papers were excluded from the analysis after reviewing the title and abstracts. After the full text review, an additional 383 papers were excluded. By assessing the literature reviews, an additional 27 articles were added. The last evaluation using the article-rating tool, with a benchmark of 70 points or higher, excluded 74 articles. Ultimately, 63 papers were included in the literature review (Figure 1) with 13 articles focused on nutrition, 17 articles focused on PA, and 33 articles focused on artificial and natural lighting, daylighting, and views.

## 3. Results

The results are divided into three sections—the nutrition, PA, and IEQ conditions specific to artificial and natural light, daylighting and views. Nutrition focused on the evidence specific to the prevention and treatment of chronic health conditions through design intervention, including complex cafeteria interventions, single interventions, and vending machines. The evidence for design intervention for physical activity influencing healthy behaviors focused on stair use and the point of decision prompts, sit to stand desks, the incentives and barriers to promote physical activity, and creating opportunity for easily taking a walk break during the day. The IEQ evidence specific to artificial lighting, natural lighting, daylighting, and views to the outside contributed to the influence of healthy behaviors contributing to positive comfort, cognitive performance, and health outcomes.

### 3.1. Nutrition

Consuming a healthy, nutritionally balanced diet can help prevent or treat many chronic health conditions, including obesity [29]. According to the 2015–2020 Dietary Guidelines for Americans (DGA), many adults need to make shifts in their current intake to achieve a healthy eating pattern. Recommended shifts include increasing the intake of whole grains, fruits, vegetables, and dairy and decreasing fat intake, in particular saturated fat [30]. These types of foods can be easily offered in a workplace cafeteria. Of note, convenience is the largest deciding factor in food selection in the work environment, with about half of employees purchasing food more than two times a week [31]. The presence of nutrition awareness services such as the provision of nutrition information, and healthy food choices being offered in cafeterias and vending machines can positively affect nutrient intake [32]. In recent years, there have been a variety of interventions employed in the workplace eating environment to improve nutritional intake. Often, the stated desired outcome of improvement, besides nutritional intake, is improvement in a health variable such as weight status, blood lipids or blood pressure. It can be difficult to tease out one intervention as the most effective as it is common that multiple interventions are implemented versus just one, i.e., choice architecture and food price incentives or nutrition education and the modification of food items offered on the menu. Choice architecture refers to the practice of influencing choice by organizing the context in which people make decisions [33]; for instance, the deliberate placement of food items available for purchase in a cafeteria or vending service [34]. Within the 13 articles that met inclusion for nutrition, six focused on the studies evaluating complex cafeteria interventions, four on single interventions and three on vending machine interventions (Appendix A).

#### 3.1.1. Complex Cafeteria Interventions

Thorndike et al. [35] conducted a two-phase food-labeling intervention that addressed low nutritional knowledge and purchasing influencers during a six month intervention in a large hospital cafeteria. Phase 1 was a simple color-coded (red, amber, green) labeling intervention of food and beverages, meant to increase nutrition knowledge. The calorie and fat content of each portion of a food or beverage was used to code the items. Red coded foods were those that should be consumed less often; those that were amber consumed in moderation; and those that were green consumed frequently. Phase 2 was the implementation of choice architecture to increase the visibility and convenience of choice of the food items considered healthy in the cafeteria. They compared the change in the sales of healthy and unhealthy items from baseline to phase 1 and from phase 1 to phase 2. During phase 1, the red item sales decreased by 9.2% and all the red beverages by 11.4%. All the green items increased by 4.5% and the green beverages increased by 9.6%. Phase 2 continued to show improvement, as the sales of red items further decreased by 4.9% and of red beverages by 11.4%, with green beverages further increasing by 4.0%. This study demonstrated the effectiveness of labeling coupled with choice architecture to improve the sales of healthy food choices [35]. 

In a similar study, Crombie et al. [36], implemented changes to menu offerings at five military cafeterias and compared customer habits to five cafeterias without the intervention, all located on one large military installation. The menu changes included the increased availability of whole grains, fruit as dessert choices, and the increased availability of lean protein options such as turkey and seafood. They also employed choice architecture. At the end of the six month study there were several positive changes in eating habits as compared to the control group. These changes included lower calorie intakes, lower total fat intake to include saturated fat, as well as the reduced intake of refined grains. Interestingly, the customers at the intervention cafeterias also reported higher customer satisfaction scores [36]. Results from both of these studies are supported by other investigators utilizing choice architecture plus food labeling to promote healthful eating at the worksite [37,38].

Demonstrating the effectiveness of combining interventions, Geaney et al. [39] conducted a cluster controlled trial in four large multinational manufacturing workplaces. The investigators compared the effectiveness of no intervention (control) to: (1) nutrition education; (2) environmental dietary modification; and (3) a combined intervention of nutrition education and environmental dietary modification. Significant positive changes included lower intakes of saturated fat and salt and increased nutrition knowledge between the baseline and follow up in the combined intervention. Small but significant changes in body weight were observed only in the combined intervention. Effects in the education and dietary modification workplace sites were smaller and generally non-significant. Of note, the worksite that had the complex workplace dietary intervention that combined nutrition education and environmental dietary modification reduced the employees’ dietary intakes of salt and saturated fat, improved their nutrition knowledge and decreased their body weight at the 7–9 month follow up. This study provides critical evidence on the effectiveness of complex workplace dietary interventions to improve eating habits, with a positive effect on body weight [39].

Lowe et al. [40] provides further evidence that combined interventions utilizing minor modifications can promote positive changes in eating. The participants were randomly assigned to one of two conditions at two workplace cafeterias. The first was environmental change, meaning that the introduction of ten new low-energy-dense foods and food labeling that provided information on energy density and macronutrient content. The second intervention consisted of the same environmental change and included pricing incentives for purchasing the low-energy dense foods and four 1 h education sessions about the benefit of consuming low-energy dense foods. Participant lunch choices were monitored electronically at the point of purchase for 3 months before the intervention was instituted and for 3 months afterward. The results indicated that the cafeteria-based intervention produced desirable reductions in energy and fat intake over the three-month period among mostly overweight and obese patrons but showed no difference when compared to the group that also offered price incentives and group education. A major potential advantage of this type of environmental intervention is that, once established, the maintenance of the intervention may be easier to achieve than the changes in nutritional intake produced by lifestyle change programs and less costly [40].

#### 3.1.2. Single Interventions

Using the menu itself as a cafeteria intervention, a group of investigators wanted to determine if changes in the menu would improve diet quality, as measured by the healthy eating index (HEI) and customer satisfaction [41]. The intervention resulted in a higher post-test HEI score (60.1 ± 8.8 points; +3.4%; *p* = 0.005) and cafeteria satisfaction scores compared with the control (49.0 ± 10.4 points; *p ˂* 0.05). Improved intervention HEI scores were attributed to changes in citrus and melon fruit (+46%), red and orange vegetables (+35%), whole grains (+181%), legumes (65%), yogurt (+45%), oils (−26%), and solid fat (−18%) consumption (*p* < 0.05) [41].

Utilizing a simple nudge design, i.e., the choice architecture in a college cafeteria setting, that consisted of changing the placement of fruit and vegetables to the beginning of the serving sequence and offering the fruit and vegetable components in separated bowls increased the self-served quantity of these items and simultaneously decreased the quantity of other the foods selected. The intervention results indicated an overall significant increase in fruit and vegetable consumption and a decrease in total energy intake. This study provides evidence that something as simple as choice architectural nudges can be effective to promote healthy eating [42].

The use of point-of-purchasing labeling in military cafeterias was evaluated by Arsenault et al. [43]. They found that 47% percent of patrons used the labels to make food choices and the label users had a significantly lower intake of fat than did non-users [43]. This study demonstrates that a low-cost easily maintained intervention can have positive effects on consumer eating habits.

Another consideration, not often mentioned, is how eating or meal consumption in the built environment is correlated to social relations when employees are offered the opportunity to share a meal with co-workers. In this context, workplace managers should consider more than nutrition and exercise in their health strategies. An attractive, comfortable dining room environment has the potential to encourage social gathering that can potentially positively affect mood, which in turn can positively affect nutrition intake [44].

These studies highlight that simple, single interventions can be utilized to promote positive changes in food consumption in the workplace environment.

#### 3.1.3. Vending Machines

Often, work environments offer food and beverage vending machine services as a stand-alone food-service option, an alternative to a cafeteria or supplemental to the cafeteria, i.e., before or after cafeteria operating hours. The options provided in these vending services can act as an incentive or barrier to healthful eating [32].

Studies utilizing the increased convenience and accessibility of vending machines to reinforce healthy food choices in individuals have been attempted less frequently in the United States as compared to Europe. Those that have been conducted have shown positive results in regard to decreasing the energy density of selected foods in both the U.S. and Europe [45,46]. The benefit of vending interventions is the control of the location presentation and type of product being sold. Product placement within the vending machine has been shown to be successful in increasing healthy choices while simultaneously decreasing unhealthy choices [46].

Recently the City of Philadelphia implemented employer-wide vending standards. The standards were designed to increase healthy snack and beverage options, to affect the sales or proportion of sales of healthy versus less healthy snacks. The sales volume and revenue for snack and beverage vending machines were monitored and evaluated after implementation and reported over two-and-a-half years. After the implementation, the proportion of sales attributable to healthy items was 40% for snacks and 46% for beverages. Healthy snack sales were 323% higher with the total snack sales reported as 17% lower. Healthy beverage sales were 33% higher, with no significant change in the total beverage sales. Interestingly, the revenue was 11% lower for snacks and 21% lower for beverages [46].

A study by Hua et al. [47] further supports vending as an avenue to encourage healthful eating. This intervention evaluated whether machines that followed healthier product guidelines regarding the foods offered, price reductions and signage would affect the sales and revenue of the vending machines. It was found that there was an interaction between healthier product guidelines and promotional signs for the vending machines and in turn increased revenue. The investigators found an overall trend to healthier purchasing, that may impact diet quality and in the long term if sustained, and positively affect health [47].

The downside of vending interventions is the reduction in sales [46]. It can be assumed that the clientele of a vending machine is looking for a snack that would be considered unhealthy and when faced with point of sale nutrition information and healthier options is likely to terminate the sale rather than select a healthier option [45,46]. This successfully reduces the caloric intake of the consumer but can negatively affect the revenue of the vendor.

### 3.2. Physical Activity

One of the modifiable risk factors for obesity prevention is PA [48]. The American College of Sports Medicine recommends that adults aim for at least 150 min of moderate-intensity cardiorespiratory exercise each week, and resistance training for each major muscle group two to three days per week using a variety of equipment and/or exercises [48]. Nationally, only one in four adults meets the federal guidelines for both aerobic and muscle-strengthening activity, with approximately 32% meeting one guideline [49]. Due to the significant amount of time that Americans spend at the workplace, this presents an opportunity to offer simple interventions to increase PA [18]. Within the 17 articles that met inclusion for PA, seven focused on stair use, three on standing or adjustable desks and seven on walking or walk breaks at work (Appendix A). 

#### 3.2.1. Stair Use and Point of Decision Prompts

Stair climbing has been suggested as a viable method for increasing PA in the public and work setting. This strategy has been given a substantial amount of attention due to its ability to impact individuals who have the potential to use stairs in their daily routine or in their workplace. Therefore, encouraging participation in stair climbing may increase physical activity in a very large demographic. In a 12 week intervention by Meyer et al. [50], which encouraged stairwells over elevator use, the participants increased their average stairwell use from 4.5 stories per day to 20.6 stories per day. In turn, the significant decreases in BMI, fat mass, waist circumference, diastolic blood pressure, low-density lipoprotein cholesterol and a significant increase in oxygen consumption (*V*o_2_), a marker of physical fitness, were observed. Interestingly, six months post intervention revealed significant decreases in blood triglycerides and insulin resistance. A qualitative study by Ruff et al. [51] found higher floor residency and BMI to be negatively related. These results suggest moderate increases in stair usage and are strongly associated with improvements in cardiorespiratory fitness and decreases in cardiovascular disease risk factors, short term and six months following the intervention. Therefore, effective strategies to increase stair frequency and volume have been investigated to further increase the levels of physical activity.

Point-of-decision prompts, stairwell visibility, and naturally lit stairwells all influence stair usage [51]. Point-of-decision prompts use text and images to encourage stair usage through messages of encouragement, motivation, and health claims of increased physical activity, representing a simple and low-cost method that has been shown to be an effective strategy of increasing stair usage [51,52,53,54]. Specifically, a systematic review by Jennings et al. [55] reported that 89% of the 52 studies analyzed used prompts including text and images, which significantly increased stair climbing. Therefore, the effectiveness of point-of-decision prompts on increasing physical activity may have a large impact on health markers. However, there appears to be caveats when analyzing the effectiveness of point-of-decision prompts. For example, Jennings and colleagues [55] also call attention to the lack of evidence of point-of-decision prompts ability to increase stair usage in the workplace when compared to public settings. Evidence shows that studies comparing stair use with escalators versus elevators more commonly reported effectiveness. Similarly, a review by Bellicha et al. [54] corroborated these findings. Therefore, there is a need for future research to study the variability in effectiveness between different environments. Other caveats when analyzing the effectiveness of point-of-decision prompts on stair usage are building occupancy, time of day, and pedestrian traffic. In a study by Olander and Eves [56], stair usage increased when there was an increase in building occupancy and when three elevators were present compared to four. On the contrary, stair usage decreased when pedestrian traffic increased and as the workday moved toward the afternoon and evening hours. These results highlight the complexity and the conditional effectiveness of prompts and stair usage. 

#### 3.2.2. Sit-to-Stand Desks

Sit-to-stand or adjustable workstations have been investigated as an effective strategy to decrease sedentary behavior and increase energy expenditure in the workplace. Specifically, oxygen consumption (*V*o_2_) and energy expenditure when standing compared to sitting are significantly higher (0.28 L/minutes versus 0.22 L/minutes; 1.36 kcal/minutes versus 1.02 kcal/minutes) [57]. Therefore, in a typical 8 h workday, this would increase the caloric expenditure by about 100–150 kcal/day. In turn, this increase in caloric expenditure can potentially have positive impacts on various health markers. This increase in energy expenditure is partially due to the increase in postural muscle engagement with standing compared to sitting. While there is a paucity of research analyzing the effectiveness of sit-to-stand workstations, relatively recent studies have highlighted the potential benefit of their presence in the workplace [57,58,59]. 

Neuhaus et al. [58] conducted a randomized control trial investigating the effectiveness of sit-to-stand workstations. The researchers found that the installation of sit-to-stand workstations decreased employee sitting time by 33 min per the 8 h workday and decreased further when combined with prompts (i.e., “Stand Up, Sit Less, Move More”) by 89 min per 8 h workday. While these findings suggest positive implications from a health and fitness perspective, other practical implications should be considered. For example, a study by Leavy and Jancey [59] investigated employee and employer perspectives on implementing sit-to-stand workstations in the workplace. Employees reported enhanced general wellbeing, workability, and practicality in response to the change in workstation design. Employers reported increased levels of staff engagement and emphasized occupational health considerations. Researchers concluded that sit-to-stand workstations were effective in breaking up prolonged sitting time, improving work performance, improving mood, and positively influencing certain health outcomes. These positive impacts on various health benefits in the workplace may encourage other healthy behaviors. Specifically, the psychological benefits associated with increased physical activity may promote other healthy lifestyle choices such as increased social interactions and increased physical activity by other means [60]. Therefore, these findings suggest sit-to-stand workstations to be effective and have important practical as well as future potential to improve health in the work setting.

#### 3.2.3. Walking during the Day (Outside or Inside)

Walking interventions in the workplace have been shown to result in significant improvements in PA levels, health perception, subjective vitality, work performance, and fatigue [61,62]. Furthermore, Puig-Ribera et al. [63] found that after 9 weeks of workplace walking (“walking while working” or utilizing “walking routes”), no significant group differences emerged. However, participants with low baseline PA levels (0–7499 steps/day) significantly increased their step count, quality of life, and work performance. Contrary results were found for those with average steps over 10,000/day with a significant decrease in number of steps. The moderately active (average steps between 7500 and 9999) saw no change. Currently, there is a dearth of evidence investigating indoor walking facilities. However, there appears to be some supportive data of outdoor walking routes/trails. A systematic review by Laine et al. [64] investigated the effectiveness and cost-effectiveness of population-level PA interventions. The newer bicycle/pedestrian trails were more likely to increase PA with a metabolic equivalent tasks (MET) of 1.843 h gained per person per day compared to others in the review. The researchers concluded that cost-effectiveness, measured as the cost-effectiveness ratio (cost per person per day divided by MET-hours gained per person per day) of these interventions were superior to the other interventions analyzed. Therefore, researchers and practitioners recommend building walking routes/trails as a useful environmental strategy to promote physical activity [65].

Furthermore, a randomized control trial by Mutrie et al. [66] investigated the effects of issuing a “Walk In to Work Out” packet that contained informative material about local cycling and walking routes among other educational content pertaining to PA. After six months, the researchers found that individuals who were given the packet were almost twice as likely to increase walking to work when compared to the control group. While extrapolation from this study must be done with caution, it suggests that knowledge about local cycle and walking routes plays a meaningful role in encouraging PA in the workplace. 

The positive implications of workplace walking on health and occupational performance have become well known to employers and employees alike. Research exploring methods of facilitating walking in the worksite have become popular as a potential solution to address PA deficiencies among working individuals. For example, “walking meetings” are proposed to increase physical activity and decrease workplace-sitting time by walking while holding meetings with colleagues. In a 3 week pilot study by Kling et al. [67], the researchers found that implementing walking meetings resulted in an increased average walking time from an average of 107 min during the baseline week to 114 min at week 2 and to 117 min at week 3. According to the authors, the intervention was easy to implement, well accepted by the white-collar workers, and successful at increasing PA levels. Incorporating walking paths/trails into the workplace design may further encourage practices that increase PA such as walking meetings.

### 3.3. IEQ-Artificial and Natural Light, Daylighting and Views

IEQ is the combined conditions inside a building. There are six primary components that contribute to the IEQ of any indoor space: (1) material composition; (2) indoor air quality (IAQ); (3) lighting and views (artificial and natural); (4) acoustics; (5) thermal comfort; and (6) occupant IEQ control [68]. Material composition influences how the selection of materials may affect the overall IEQ of the space and therefore, the occupants. IAQ data provide a baseline for identifying possible environmental hazards. Lighting and acoustic data support the theory of interaction between factors within the space (i.e., natural light streaming through a window creating a material response effecting the eye; and increased sound levels effecting occupants’ stress and risk of work errors). Thermal comfort is that condition of mind which expresses satisfaction with the thermal environment [69]. Thermal comfort occurs when body temperatures are held within narrow ranges, skin moisture is low, and the effort to mentally maintain comfort is minimal. Environmentally, common measurements include ambient temperature, relative humidity, air movement, and carbon dioxide (CO_2_) levels. The occupant control of the indoor environment provides a manipulation of the IAQ, lighting, acoustics, and thermal comfort using individualized controls and mechanical or computerized overrides for environmental controls such as sensors, shading, light levels, and temperature.

The articles included in this review are limited to artificial lighting, natural lighting, daylighting, and views to the outside aimed at developing evidence-based design applications supporting behavior modification to improve measurable health outcomes. Artificial lighting is any light source that is produced by manmade, typically electrical, means. Natural light is the light that comes from the sun. Daylighting uses natural light as a substitute for artificial lighting [69]. Successful daylighting strategies use daylight for basic ambient light, utilizing artificial lighting to supplement lighting needs. One of the primary goals for daylighting is the reduction of electricity, which may have a positive impact on facility costs. Views to the outside is separate from access to natural light, providing occupants with visual distraction. 

Within the 33 articles that met inclusion for IEQ, 17 centered around studies evaluating health outcomes related to circadian disruption. Six articles focused on the effect of views and natural light and the relationship to health outcomes. The remaining 10 articles addressed artificial light, including the type, color rendering, amount, and glare with outcomes related to comfort, fatigue, alertness, and cognitive performance (Appendix A).

#### 3.3.1. Circadian Disruption

Circadian rhythms are the biological cycles that regulate the sleep-wake cycle, repeating about every 24 h [70]. Humans have become increasingly active during the late evening hours, shifting from a primarily diurnal lifestyle to a more nocturnal one, leading to circadian disruption at the system, tissue and cellular levels [71]. Circadian disruption is associated with metabolic disturbances including obesity, metabolic syndrome, diabetes, cardiovascular disease, cancer, and other physical and mental disorders [72,73,74]. 

Bright light exposure at night through artificial light sources is commonplace and has contributed to a 24 h society in many well developed countries, allowing for an extended workday or a separate night shift. The extended day influences social opportunities where individuals are choosing exposure to light at night during the sleep-wake cycles of the circadian rhythms [75]. Exposure to artificial light at night (ALAN) has effectively reduced the exposure to dark at night, affecting a disruption of the biological clock and the suppression of melatonin production [71]. A systematic literature review evaluated the studies on artificial light and found that light intensity, exposure duration, timing, wavelength, individual light habits, and outdoor ALAN levels were factors related to ALAN exposure conditions [71]. 

An evaluation of a series of meta-analyses and systemic literature reviews published between 2005 and 2017 found significant associations between shift work and health outcomes [71]. The studies represented multiple occupational settings and focused on the health outcomes associated with circadian disruption. The researchers concluded that there was evidence linking non-standard work time with the increased risk of negative health outcomes. The evidence presented supported the findings for the increased risk of cardiovascular disease [76], metabolic syndrome [77], type-2 diabetes [78,79], breast cancer [80,81,82,83,84,85,86], prostate cancer [87], colorectal cancer [88], early miscarriage [89], and depression [90].

#### 3.3.2. Natural Light and Views

The amount of exposure to natural lighting and daylighting has been linked to various mental and physical health impacts of building occupants. One study found that workers in environments with windows had increased exposure to light in the workplace and showed a trend toward more physical activity, longer sleep duration, better sleep quality, and an improved quality of life compared to workers in windowless environments [89]. Furthermore, windows and daylight provide micro-restorative effects by lowering blood pressure, increasing oxygen saturation, and improving circadian rhythms among registered nurses in acute-care units [91]. 

The impact of windows in the indoor environment is two-fold by allowing occupants visual access to natural elements and natural lighting from the sun. A survey of office employees examined the impact of natural elements and sunlight exposure on mental health and work attitudes [91]. The study found that natural elements and sunlight exposure were positively related to job satisfaction and organizational commitment, and reduced depression and anxiety. Nature and daylight exposure were positively related to energy levels and were found to be beneficial to affect a reduction in stress levels [91]. 

Various work environments, indoor and outdoor, may have beneficial or harmful effects on the physical and mental health of employees due to the levels of natural light experienced throughout the year [92,93]. For example, one study found that indoor, outdoor, and night workers all experienced varying exposures to light throughout the year which was associated with varying levels of wellbeing and mood [94]. The results revealed that night workers experienced levels of light exposure expected to suppress melatonin and initiate the circadian phase shift; indoor workers experienced low light intensities associated with reduced mood and wellbeing; and during summer hours, outdoor workers experienced light levels comparable to light therapy for the treatment of depression [94]. Manipulations of light systems may provide exposure times and intensities to minimize melatonin suppression, influencing passive compliance that influences circadian rhythms to reduce risks effecting mood and wellbeing. 

#### 3.3.3. Artificial Light

Artificial light is composed of visible light, ultraviolet (UV) light and infrared (IR) light. Artificial light contributes to the quality of life but may influence negative outcomes based on exposure. 

The circadian rhythm is affected by the exposure times and light level variations throughout the day. A study of hospitalized adults found that this cohort experienced exposure to low light levels, 24 h per day, suggesting a lack of fluctuation between varying light levels to maintain normal circadian rhythms [95]. With higher light exposure, patients reported less fatigue and lower total mood disturbance in participants with pain. These findings suggest that light exposures may provide psychophysical benefits. Additional research is needed to determine if manipulating light exposure for hospital patients would increase the quality of sleep and mood while reducing the severity of pain. Another study found that office workers receiving high levels of circadian effective light in the morning experienced reduced sleep onset latency, increased circadian entrainment, increased sleep quality, and reduced depression [96]. Designing for health outcomes may include the light systems specified for improving circadian rhythm entrainment. 

The impact of variable lighting conditions on circadian rhythms and subjective mood was also examined by exposing two groups of employees to different lighting environments in otherwise constant work environments [96]. One room of occupants experienced exposure to normal lighting of illuminance level (500 lx) and a color temperature of 4000 K, while the second room of occupants experienced exposure to variable illuminance levels (500–1800 lx) with an increased color temperature of 6500 K. For the variable environment, illuminance levels were gradually altered throughout the day at specific time intervals. The results indicated a potential benefit of exposure to variable lighting and an associated increase in the mood dimensions of “activity” compared to increased mood dimensions of “deactivation” and “fatigue” associated with regular lighting conditions. 

Artificial light type and color rendering also have various effects on building occupants. The exposure to variations of artificial illuminance levels and color temperatures may show a beneficial effect on work performance, visual comfort, alertness, and job satisfaction [97]. The effects of light-emitting diode (LED), compact fluorescent (FLcomp), fluorescent with warm color temperature (FLwarm), and cool color temperture (FLcool) were compared based on the student participants’ performance, alertness, visual comfort level, and light preferences. The study concluded that human performance improved under cold color temperatures when working with fluorescent and LED lamps of the same color temperatures [97]. Similarly, a study examined the effect of increased levels of illuminance on subjective measures, sustained attention, cognitive performance, and physiological daytime arousal to further determine the alertness and vitality impacts of the light on office workers finding that participants reported feeling less sleepy and more energetic; with shorter reaction times; and increased physiological arousal when exposed to the higher lighting condition [98].

LED task lighting and discomfort, eye fatigue, the perception of job contentment, usability, and musculoskeletal discomfort among office workers was examined using a baseline (no LED task light) versus intervention study (LED adjustable task light provided). The use of LED task lights resulted in significant improvements on the reported level of discomfort, eye fatigue, perception of job contentment and posture [99]. Other aspects of artificial lighting were found to contribute to dissatisfaction with results revealing that panel heights, high reflected glare on computer screens, desktop illuminances outside of 300–500 lx, desktop illuminance uniformity less than 0.5, and being in a workstation distant from a window were all associated with a higher levels of dissatisfaction with lighting [100]. 

Lastly, the effects of visual acuity and the control of lighting were considered. Satisfaction with the workplace may be influenced by the control of the environment. A study investigated the perceptions of the lighting environment and the levels of satisfaction in the workplace, due to variations in lighting among medical-surgical nurses and found a significant relationship between nurse access to lighting controls and satisfaction about the lighting environment [101]. Demographic variables, specifically age, may also play a role in the effect of lighting conditions on work performance. A literature review detailed the age-related changes in visual and non-visual functions among older-age workers and the relationship between light and work performance. The review concluded that eye changes and ocular disease in older-age people may have an impact on performance and overall wellbeing [102]. The results suggest that work conditions, particularly lighting, must consider the age of the workers exposed. 

## 4. Discussion

Nightingale’s environmental theory, supported by the theory of integration utilizing the transactional model, supports the human-environment relationship, provides a framework for research focused on improving overall public health with an emphases on healthy nutrition, increased physical activity, and the reduction of obesity through environmental influence for behavior modification. 

This inquiry identified elements of the built environment related to the exposure to light and views that affect human behavior related to nutrition and physical activity. This review highlights the interconnections between the elements of nutrition, PA and IEQ to show the significance of designing the built environment to support complete health.

Improving nutrition intake in the workplace is crucial, as a recent cross-sectional study of over 5000 employed adults found that almost a quarter of this sample obtain foods at work during the week, averaging over 1200 calories per person with food choices being high in fat, added sugars, or sodium and low in whole grains and fruit [103]. Nutrition interventions to improve the quality of food intake in the work environment have employed a variety of simple [41,42,43] to complex strategies [35,36,37,38,39,40]. A key theme throughout is that the eating venue must make it easy, i.e., routine or habit forming, for the employee to choose healthy food options. 

Improvements in workplace eating venues, whether cafeterias or vending machines, have the potential over the long term to improve the overall diet quality of the workplace occupants in the context of reducing total calorie intake, as well as the consumption of fat, saturated fat, refined grains, and salt. There is also the potential to increase the consumption of fruit, vegetables, and whole grains which are food groups of concern in the U.S. diet, a method to maintain or reduce body weight and improve diet quality, both important risk factors for type 2 diabetes mellitus and other chronic disease conditions [30]. In a similar fashion, modifying lighting in the work environment can have a positive effect on sleep quality [104] and quantity [96] that has been correlated with a beneficial effect on food intake [70,71]. Natural light exposure [91,92], increased light exposure [104], circadian effective light in the morning [96], and variable lightning [105] can enhance the mood reduce depression, that can also positively influence diet [70,72]. 

In addition to good nutritional intake, physical activity plays an important role in overall health. A study of employed adults found that nearly a third were obese and engaged in less frequent leisure time physical activity than normal-weight adults [106]. Encouraging PA during the workday is a viable method to increase overall PA throughout the week, with corresponding positive effects on several health variables that include body weight, other markers of metabolic disease and mood. Interventions that encourage PA at work range from stair use [51,52,53,54,55,56,57], sit/stand furniture [57,58,59], and walking [61,62,63,64,66]. Furthermore, the psychological benefits associated with increased physical activity may promote other healthy lifestyle choices such as increased social interactions, increased physical activity by other means [60], and improvements in food intake [72]. Highlighting the complexity of factors influencing behavior, natural light exposure has also been shown to increase PA, improve the quality of life and mood and even lower blood pressure [92,104].

Lighting can contribute in a positive way to support health, either directly or indirectly. However, ALAN may contribute a negative influence. As humans have become more active at night, using artificial lighting to supplement a longer day has increased the incidence of circadian disruption leading to increased obesity, diabetes, metabolic syndrome, cardiovascular disease, cancer, and other medical maladies including mental and physical disorders [72,73,74]. Intentional design to provide appropriate exposure to light during the workday may improve circadian entrainment. One system to consider is the circadian rhythm lighting system. A variable lighting system or circadian system have been found to increase entrainment, increase sleep quality, and reduce depression [96]. From a behavioral perspective, the awareness and intentional behaviors to improve circadian entrainment can mitigate the negative effects of ALAN. Nutrition intake and PA are important factors. Increased nutrition, weight loss, and reduced salt and saturated fat have influenced cancers, including breast, prostate, and colorectal. The use of labeling, choice architecture and education can improve nutritional intake and increase physical activity, choices that are controlled by the individual that improve physical, mental, and behavioral health. 

Evidence on the use of architecture and the built environment to encourage the behaviors associated with improved health has not previously been collated to determine the overall impact on health. This study evaluated peer-reviewed research to bring together the science, technology and behavior modification strategies for improving health, specifically in the workplace. Within the built environment, IEQ factors are interrelated, contributing to the overall quality of the IEQ of the building and affecting the occupants. One of the primary components of IEQ is lighting (natural and artificial), including daylight and views which play a role in influencing behavior through active choices and passive compliance. Through design, both passive compliance and active behavior modification may produce positive changes that influence occupant outcomes. 

This review highlights the research published in the scientific literature presenting evidence to support behavior modification in the workplace for improved health. Anecdotally, the inclusion of breakrooms with refrigeration and the provision of a microwave may provide the opportunity for individuals to bring healthy foods from home by choice, however, there was a dearth of evidence to support this common-sense idea. Similarly, the evidence from the IEQ studies present strong recommendations for the improvement of the IEQ in the workplace, however, there is limited evidence for the implementation of strategies published. This study did not include or evaluate municipal health promotion programs regarding nutrition or physical activity. Data from those programs may provide insight to conclusions not made in this review. 

## 5. Conclusions

Designing to improve and increase behaviors leading to healthy choices for nutrition and physical activity is a combination of the design of the environment and intentional programming, making it easy, routine and habit forming. The intersection or combination of effects of the elements reviewed greatly enhance the opportunity for building occupants to improve their health. Promotional signage, choice architecture and the availability of healthy food options are all programmatic interventions that encourage nutritional improvement. Similarly, physical activity at work is a conscious choice. The design of monumental stairs or fire stairs with glazing to provide natural light entices occupants to take the stairs rather than waiting on the elevator. Natural light or the use of variable lighting systems may contribute to health and wellbeing through passive compliance, encouraging physical activity and improving psycho-social and physical occupant outcomes. Improvement in lighting technologies offers solutions intended to reduce circadian disruption, minimizing the risk of negative health outcomes including cancers, metabolic syndrome, and cardiovascular events. Design that encourages healthy behaviors within the workplace advances the philosophy of design for health.

## Figures and Tables

**Figure 1 ijerph-17-05049-f001:**
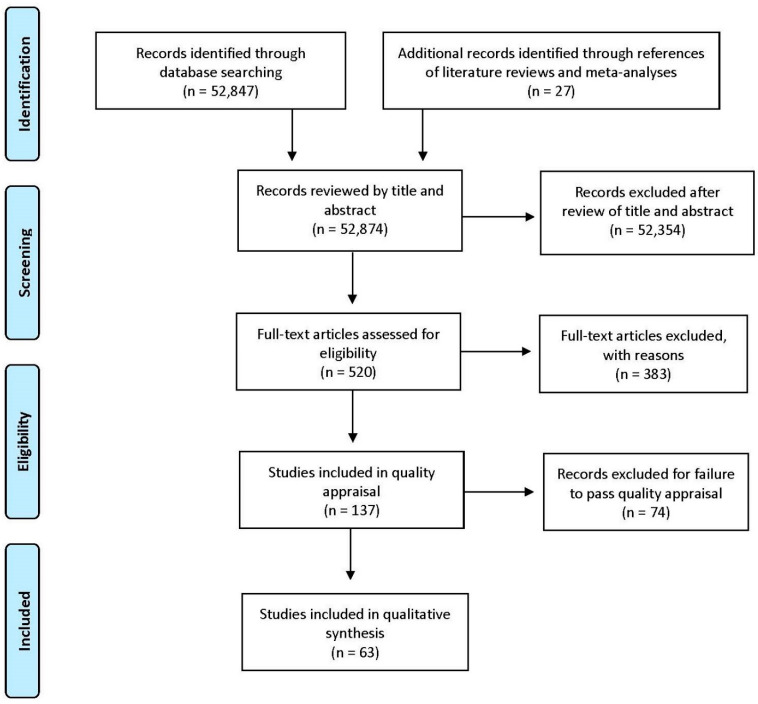
Literature search results [24,25,26,27].

**Table 1 ijerph-17-05049-t001:** Search terms for nutrition, physical activity, and natural and artificial lighting.

Absenteeism	Academic	Adjustable workstations
Ambient light	Building	Building outcomes
Cafeteria environment	Choice architecture	Circadian rhythm
Comfort	Control	Convenience foods
Corporate	Cost	Daylight
Diet	Diet quality	Dining facility
Direct light	Energy consumption	Energy density
Environment	Environmental intervention	Facilitators and barriers
Fenestration	Food labeling	Glare
Government	Health	Health promotion
Healthcare	Healthy eating index	Human performance
Illumination	Indirect light	Industrial
Integrative technologies	Light	Light level
Light sensor	Lighting	Lighting level
Long-term Care	Mental health	Morale
Motion sensor	Natural light	Nutrition
Nutrition policy	Obesity	Occupancy sensors
Occupant outcomes	Occupation	Office
Operable windows	Outdoor lighting	Personal control
Photometric sensor	Physical activity	Presenteeism
Productivity	Retail	Satisfaction
School	Security lighting	Shading
Shadows	Sit to stand	Stair prompts
Stairs	Surface reflectance	Task lights
Vending machines	Views	Visual comfort
Visual tasks	Walking paths	Weather
Windows	Workplace	Workspace

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
