# Peer review of "Healthy Behaviors through Behavioral Design–Obesity Prevention"

_ijerph, 2020, doi:10.3390/ijerph17145049_

Round 1

Reviewer 1 Report

Several comments could be considered in order to improve the quality of this paper.

  1. I feel very confused on the core topic. According to the title, it look like that this review will investigate the link between human behaviour and IEQ factors and obesity. But, when reading the section 3.3, the impact of IEQ on general health problems of human has become the topic. Why?
  2. Method and materials: this part may need more explanations according to a normal systematic review way.

1) It would be good if a flow chart and more words can be applied to explain the peer-reviewed rating method when selecting the articles.

2)  In the Figure1, Full-text articles excluded with reasons (n=383), what were the reasons?

3) What were the inclusive and exclusive criteria for literature searching?

  1. For the IEQ literature searching, from my point of view, Daylighting = natural lighting. It could be hard to understand why two definitions were applied in the literature searching and review. This could be a problem with the unclear definitions.
  2. It can be understood that the nutrition and Physical Activity can take significant effects on the obesity (mentioned in the section 3.1 & 3.2). However, only simple research activities were presented in the two parts. It would be good if more deep analysis can be presented.
  3. Why the authors only selected the artificial lighting, natural lighting, daylighting, and views to the outside as the key IEQ factors, even six big types of factors were presented?

The choice of IEQ factors could be hard to accept since other aspects can significantly affect the human health (including obesity).

  1. For the section 3. 3 IEQ, no clear literatures can be found in terms of obesity. Why?

In general, the organization of this review is not proper. More proper literatures may still need to improve the reviewing, especially for the topic of Obesity.

Reviewer 2 Report

The manuscript “Healthy Behaviors Through Behavioral Design-Obesity Prevention” by Funderburk et al. reviewed how the behavioral modification of nutrition, physical activity, and indoor environmental quality could be used in the prevention of obesity and related comorbidities.

The manuscript accomplishes a well-designed review of the existing literature on the subject. Nonetheless, minor changes could be made to enhance this manuscript to a higher scientific level. Thus, this manuscript is not ready enough to be published in Int. J. Environ. Res. Public Health from MDPI.

You can find my comments below:

  • The authors could include a table summarizing the information extracted from the papers (68) included in the review. Tables should include statistical information, year of publication, number of people that were included in the study, and minor details that could help with the understanding of the main results and conclusions of the paper. Hence, each table could serve as a brief summary of the research.
  • There are minor errors in the use of abbreviations: the authors use min vs. minutes, hours, QoL (quality of life), among others. They should be consistent and include abbreviations only if the term is used more than once. Oxygen consumption should be written VO2.

Round 2

Reviewer 1 Report

My comments have been addressed.